# Simple and Scalable Predictive Uncertainty Estimation using Deep Ensembles

**Balaji Lakshminarayanan**     **Alexander Pritzel**     **Charles Blundell**
DeepMind
{balajiln,apritzel,cblundell}@google.com

## Abstract

Deep neural networks (NNs) are powerful black box predictors that have recently achieved impressive performance on a wide spectrum of tasks. Quantifying predictive uncertainty in NNs is a challenging and yet unsolved problem. Bayesian NNs, which learn a distribution over weights, are currently the state-of-the-art for estimating predictive uncertainty; however these require significant modifications to the training procedure and are computationally expensive compared to standard (non-Bayesian) NNs. We propose an alternative to Bayesian NNs that is simple to implement, readily parallelizable, requires very little hyperparameter tuning, and yields high quality predictive uncertainty estimates. Through a series of experiments on classification and regression benchmarks, we demonstrate that our method produces well-calibrated uncertainty estimates which are as good or better than approximate Bayesian NNs. To assess robustness to dataset shift, we evaluate the predictive uncertainty on test examples from known and unknown distributions, and show that our method is able to express higher uncertainty on out-of-distribution examples. We demonstrate the scalability of our method by evaluating predictive uncertainty estimates on ImageNet.

## 1   Introduction

Deep neural networks (NNs) have achieved state-of-the-art performance on a wide variety of machine learning tasks [35] and are becoming increasingly popular in domains such as computer vision [32], speech recognition [25], natural language processing [42], and bioinformatics [2, 61]. Despite impressive accuracies in supervised learning benchmarks, NNs are poor at quantifying predictive uncertainty, and tend to produce overconfident predictions. Overconfident incorrect predictions can be harmful or offensive [3], hence proper uncertainty quantification is crucial for practical applications.

Evaluating the quality of predictive uncertainties is challenging as the 'ground truth' uncertainty estimates are usually not available. In this work, we shall focus upon two evaluation measures that are motivated by practical applications of NNs. Firstly, we shall examine *calibration* [12, 13], a frequentist notion of uncertainty which measures the discrepancy between subjective forecasts and (empirical) long-run frequencies. The quality of calibration can be measured by *proper scoring rules* [17] such as log predictive probabilities and the Brier score [9]. Note that calibration is an orthogonal concern to accuracy: a network's predictions may be accurate and yet miscalibrated, and vice versa. The second notion of quality of predictive uncertainty we consider concerns generalization of the predictive uncertainty to domain shift (also referred to as *out-of-distribution* examples [23]), that is, measuring if the network *knows what it knows*. For example, if a network trained on one dataset is evaluated on a completely different dataset, then the network should output high predictive uncertainty as inputs from a different dataset would be far away from the training data. Well-calibrated predictions that are robust to model misspecification and dataset shift have a number of important practical uses (e.g., weather forecasting, medical diagnosis).

There has been a lot of recent interest in adapting NNs to encompass uncertainty and probabilistic methods. The majority of this work revolves around a Bayesian formalism [4], whereby a prior distribution is specified upon the parameters of a NN and then, given the training data, the posterior distribution over the parameters is computed, which is used to quantify predictive uncertainty. Since exact Bayesian inference is computationally intractable for NNs, a variety of approximations have been developed including Laplace approximation [40], Markov chain Monte Carlo (MCMC) methods [46], as well as recent work on variational Bayesian methods [6, 19, 39], assumed density filtering [24], expectation propagation [21, 38] and stochastic gradient MCMC variants such as Langevin diffusion methods [30, 59] and Hamiltonian methods [53]. The quality of predictive uncertainty obtained using Bayesian NNs crucially depends on (i) the degree of approximation due to computational constraints and (ii) *if* the prior distribution is 'correct', as priors of convenience can lead to unreasonable predictive uncertainties [50]. In practice, Bayesian NNs are often harder to implement and computationally slower to train compared to non-Bayesian NNs, which raises the need for a 'general purpose solution' that can deliver high-quality uncertainty estimates and yet requires only minor modifications to the standard training pipeline.

Recently, Gal and Ghahramani [15] proposed using *Monte Carlo dropout* (MC-dropout) to estimate predictive uncertainty by using *Dropout* [54] at test time. There has been work on approximate Bayesian interpretation [15, 29, 41] of dropout. MC-dropout is relatively simple to implement leading to its popularity in practice. Interestingly, dropout may also be interpreted as *ensemble model combination* [54] where the predictions are averaged over an ensemble of NNs (with parameter sharing). The ensemble interpretation seems more plausible particularly in the scenario where the dropout rates are not tuned based on the training data, since any sensible approximation to the true Bayesian posterior distribution has to depend on the training data. This interpretation motivates the investigation of ensembles as an alternative solution for estimating predictive uncertainty.

It has long been observed that ensembles of models improve predictive performance (see [14] for a review). However it is not obvious when and why an ensemble of NNs can be expected to produce good uncertainty estimates. Bayesian model averaging (BMA) assumes that the true model lies within the hypothesis class of the prior, and performs *soft model selection* to find the single best model within the hypothesis class [43]. In contrast, ensembles perform *model combination*, i.e. they combine the models to obtain a more powerful model; ensembles can be expected to be better when the true model does not lie within the hypothesis class. We refer to [11, 43] and [34, §2.5] for related discussions. It is important to note that even exact BMA is not guaranteed be robust to mis-specification with respect to domain shift.

*Summary of contributions*: Our contribution in this paper is two fold. First, we describe a simple and scalable method for estimating predictive uncertainty estimates from NNs. We argue for training probabilistic NNs (that model predictive distributions) using a proper scoring rule as the training criteria. We additionally investigate the effect of two modifications to the training pipeline, namely (i) *ensembles* and (ii) *adversarial training* [18] and describe how they can produce smooth predictive estimates. Secondly, we propose a series of tasks for evaluating the quality of the predictive uncertainty estimates, in terms of calibration and generalization to unknown classes in supervised learning problems. We show that our method significantly outperforms (or matches) MC-dropout. These tasks, along with our simple yet strong baseline, serve as an useful benchmark for comparing predictive uncertainty estimates obtained using different Bayesian/non-Bayesian/hybrid methods.

*Novelty and Significance*: Ensembles of NNs, or *deep ensembles* for short, have been successfully used to boost predictive performance (e.g. classification accuracy in ImageNet or Kaggle contests) and adversarial training has been used to improve robustness to adversarial examples. However, to the best of our knowledge, ours is the first work to investigate their usefulness for predictive uncertainty estimation and compare their performance to current state-of-the-art approximate Bayesian methods on a series of classification and regression benchmark datasets. Compared to Bayesian NNs (e.g. variational inference or MCMC methods), our method is much simpler to implement, requires surprisingly few modifications to standard NNs, and well suited for distributed computation, thereby making it attractive for large-scale deep learning applications. To demonstrate scalability of our method, we evaluate predictive uncertainty on ImageNet (and are the first to do so, to the best of our knowledge). Most work on uncertainty in deep learning focuses on Bayesian deep learning; we hope that the simplicity and strong empirical performance of our approach will spark more interest in non-Bayesian approaches for predictive uncertainty estimation.

# 2 Deep Ensembles: A Simple Recipe For Predictive Uncertainty Estimation

## 2.1 Problem setup and High-level summary

We assume that the training dataset $\mathcal{D}$ consists of $N$ i.i.d. data points $\mathcal{D} = \{\mathbf{x}_n, y_n\}_{n=1}^N$, where $\mathbf{x} \in \mathbb{R}^D$ represents the $D$-dimensional features. For classification problems, the label is assumed to be one of $K$ classes, that is $y \in \{1, \ldots, K\}$. For regression problems, the label is assumed to be real-valued, that is $y \in \mathbb{R}$. Given the input features $\mathbf{x}$, we use a neural network to model the probabilistic predictive distribution $p_\theta(y|\mathbf{x})$ over the labels, where $\theta$ are the parameters of the NN.

We suggest a simple recipe: (1) use a proper scoring rule as the training criterion, (2) use *adversarial training* [18] to smooth the predictive distributions, and (3) train an *ensemble*. Let $M$ denote the number of NNs in the ensemble and $\{\theta_m\}_{m=1}^M$ denote the parameters of the ensemble. We first describe how to train a single neural net and then explain how to train an ensemble of NNs.

## 2.2 Proper scoring rules

Scoring rules measure the quality of predictive uncertainty (see [17] for a review). A scoring rule assigns a numerical score to a predictive distribution $p_\theta(y|\mathbf{x})$, rewarding better calibrated predictions over worse. We shall consider scoring rules where a higher numerical score is better. Let a scoring rule be a function $S(p_\theta, (y, \mathbf{x}))$ that evaluates the quality of the predictive distribution $p_\theta(y|\mathbf{x})$ relative to an event $y|\mathbf{x} \sim q(y|\mathbf{x})$ where $q(y, \mathbf{x})$ denotes the true distribution on $(y, \mathbf{x})$-tuples. The expected scoring rule is then $S(p_\theta, q) = \int q(y, \mathbf{x}) S(p_\theta, (y, \mathbf{x})) dy d\mathbf{x}$. A *proper scoring rule* is one where $S(p_\theta, q) \leq S(q, q)$ with equality if and only if $p_\theta(y|\mathbf{x}) = q(y|\mathbf{x})$, for all $p_\theta$ and $q$. NNs can then be trained according to measure that encourages calibration of predictive uncertainty by minimizing the loss $\mathcal{L}(\theta) = -S(p_\theta, q)$.

It turns out many common NN loss functions are proper scoring rules. For example, when maximizing likelihood, the score function is $S(p_\theta, (y, \mathbf{x})) = \log p_\theta(y|\mathbf{x})$, and this is a proper scoring rule due to Gibbs inequality: $S(p_\theta, q) = \mathbb{E}_{q(\mathbf{x})} q(y|\mathbf{x}) \log p_\theta(y|\mathbf{x}) \leq \mathbb{E}_{q(\mathbf{x})} q(y|\mathbf{x}) \log q(y|\mathbf{x})$. In the case of multi-class $K$-way classification, the popular softmax cross entropy loss is equivalent to the log likelihood and is a proper scoring rule. Interestingly, $\mathcal{L}(\theta) = -S(p_\theta, (y, \mathbf{x})) = K^{-1} \sum_{k=1}^K \big(\delta_{k=y} - p_\theta(y = k|\mathbf{x})\big)^2$, i.e., minimizing the squared error between the predictive probability of a label and one-hot encoding of the correct label, is also a proper scoring rule known as the Brier score [9]. This provides justification for this common trick for training NNs by minimizing the squared error between a binary label and its associated probability and shows it is, in fact, a well defined loss with desirable properties.[1]

### 2.2.1 Training criterion for regression

For regression problems, NNs usually output a single value say $\mu(\mathbf{x})$ and the parameters are optimized to minimize the mean squared error (MSE) on the training set, given by $\sum_{n=1}^N \big(y_n - \mu(\mathbf{x}_n)\big)^2$. However, the MSE does not capture predictive uncertainty. Following [47], we use a network that outputs two values in the final layer, corresponding to the predicted mean $\mu(\mathbf{x})$ and variance[2] $\sigma^2(\mathbf{x}) > 0$. By treating the observed value as a sample from a (heteroscedastic) Gaussian distribution with the predicted mean and variance, we minimize the negative log-likelihood criterion:

$$-\log p_\theta(y_n|\mathbf{x}_n) = \frac{\log \sigma_\theta^2(\mathbf{x})}{2} + \frac{\big(y - \mu_\theta(\mathbf{x})\big)^2}{2\sigma_\theta^2(\mathbf{x})} + \text{constant}. \tag{1}$$

We found the above to perform satisfactorily in our experiments. However, two simple extensions are worth further investigation: (i) Maximum likelihood estimation over $\mu_\theta(\mathbf{x})$ and $\sigma_\theta^2(\mathbf{x})$ might overfit; one could impose a prior and perform maximum-a-posteriori (MAP) estimation. (ii) In cases where the Gaussian is too-restrictive, one could use a complex distribution e.g. mixture density network [5] or a heavy-tailed distribution.

## 2.3 Adversarial training to smooth predictive distributions

Adversarial examples, proposed by Szegedy et al. [55] and extended by Goodfellow et al. [18], are those which are 'close' to the original training examples (e.g. an image that is visually indistinguishable from the original image to humans), but are misclassified by the NN. Goodfellow et al. [18] proposed the *fast gradient sign method* as a fast solution to generate adversarial examples. Given an input $\mathbf{x}$ with target $y$, and loss $\ell(\theta, \mathbf{x}, y)$ (e.g. $-\log p_\theta(y|\mathbf{x})$), the fast gradient sign method generates an adversarial example as $\mathbf{x}' = \mathbf{x} + \epsilon \operatorname{sign}(\nabla_\mathbf{x} \ell(\theta, \mathbf{x}, y))$, where $\epsilon$ is a small value such that the max-norm of the perturbation is bounded. Intuitively, the adversarial perturbation creates a new training example by adding a perturbation along a direction which the network is likely to increase the loss. Assuming $\epsilon$ is small enough, these adversarial examples can be used to augment the original training set by treating $(\mathbf{x}', y)$ as additional training examples. This procedure, referred to as *adversarial training*,[3] was found to improve the classifier's robustness [18].

Interestingly, adversarial training can be interpreted as a computationally efficient solution to smooth the predictive distributions by increasing the likelihood of the target around an $\epsilon$-neighborhood of the observed training examples. Ideally one would want to smooth the predictive distributions along all $2^D$ directions in $\{1, -1\}^D$; however this is computationally expensive. A random direction might not necessarily increase the loss; however, adversarial training by definition computes the direction where the loss is high and hence is better than a random direction for smoothing predictive distributions. Miyato et al. [44] proposed a related idea called *virtual adversarial training* (VAT), where they picked $\Delta\mathbf{x} = \arg\max_{\Delta\mathbf{x}} \operatorname{KL}(p(y|\mathbf{x})||p(y|\mathbf{x} + \Delta\mathbf{x}))$; the advantage of VAT is that it does not require knowledge of the true target $y$ and hence can be applied to semi-supervised learning. Miyato et al. [44] showed that distributional smoothing using VAT is beneficial for efficient semi-supervised learning; in contrast, we investigate the use of adversarial training for predictive uncertainty estimation. Hence, our contributions are complementary; one could use VAT or other forms of adversarial training, cf. [33], for improving predictive uncertainty in the semi-supervised setting as well.

## 2.4 Ensembles: training and prediction

The most popular ensembles use decision trees as the base learners and a wide variety of method have been explored in the literature on ensembles. Broadly, there are two classes of ensembles: *randomization*-based approaches such as random forests [8], where the ensemble members can be trained in parallel without any interaction, and *boosting*-based approaches where the ensemble members are fit sequentially. We focus only on the randomization based approach as it is better suited for distributed, parallel computation. Breiman [8] showed that the generalization error of random forests can be upper bounded by a function of the strength and correlation between individual trees; hence it is desirable to use a *randomization scheme* that de-correlates the predictions of the individual models as well as ensures that the individual models are strong (e.g. high accuracy). One of the popular strategies is *bagging* (a.k.a. bootstrapping), where ensemble members are trained on different bootstrap samples of the original training set. If the base learner lacks intrinsic randomization (e.g. it can be trained efficiently by solving a convex optimization problem), bagging is a good mechanism for inducing diversity. However, if the underlying base learner has multiple local optima, as is the case typically with NNs, the bootstrap can sometimes hurt performance since a base learner trained on a bootstrap sample sees only 63% unique data points.[4] In the literature on decision tree ensembles, Breiman [8] proposed to use a combination of bagging [7] and random subset selection of features at each node. Geurts et al. [16] later showed that bagging is unnecessary if additional randomness can be injected into the random subset selection procedure. Intuitively, using more data for training the base learners helps reduce their bias and ensembling helps reduce the variance.

We used the entire training dataset to train each network since deep NNs typically perform better with more data, although it is straightforward to use a random subsample if need be. We found that random initialization of the NN parameters, along with random shuffling of the data points, was sufficient to obtain good performance in practice. We observed that bagging deteriorated performance in our experiments. Lee et al. [36] independently observed that training on entire dataset with random initialization was better than bagging for deep ensembles, however their goal was to improve

predictive accuracy and not predictive uncertainty. The overall training procedure is summarized in Algorithm 1.

---

**Algorithm 1** Pseudocode of the training procedure for our method

---

1: ▷ *Let each neural network parametrize a distribution over the outputs, i.e. $p_\theta(y|\mathbf{x})$. Use a proper scoring rule as the training criterion $\ell(\theta, \mathbf{x}, y)$. Recommended default values are $M = 5$ and $\epsilon = 1\%$ of the input range of the corresponding dimension (e.g 2.55 if input range is [0,255]).*
2: Initialize $\theta_1, \theta_2, \ldots, \theta_M$ randomly
3: **for** $m = 1 : M$ **do**                            ▷ *train networks independently in parallel*
4:    Sample data point $n_m$ randomly for each net    ▷ *single $n_m$ for clarity, minibatch in practice*
5:    Generate adversarial example using $\mathbf{x}'_{n_m} = \mathbf{x}_{n_m} + \epsilon \, \text{sign}\big(\nabla_{\mathbf{x}_{n_m}} \ell(\theta_m, \mathbf{x}_{n_m}, y_{n_m})\big)$
6:    Minimize $\ell(\theta_m, \mathbf{x}_{n_m}, y_{n_m}) + \ell(\theta_m, \mathbf{x}'_{n_m}, y_{n_m})$ w.r.t. $\theta_m$      ▷ *adversarial training (optional)*

---

We treat the ensemble as a uniformly-weighted mixture model and combine the predictions as $p(y|\mathbf{x}) = M^{-1} \sum_{m=1}^{M} p_{\theta_m}(y|\mathbf{x}, \theta_m)$. For classification, this corresponds to averaging the predicted probabilities. For regression, the prediction is a mixture of Gaussian distributions. For ease of computing quantiles and predictive probabilities, we further approximate the ensemble prediction as a Gaussian whose mean and variance are respectively the mean and variance of the mixture. The mean and variance of a mixture $M^{-1} \sum \mathcal{N}\big(\mu_{\theta_m}(\mathbf{x}), \sigma^2_{\theta_m}(\mathbf{x})\big)$ are given by $\mu_*(\mathbf{x}) = M^{-1} \sum_m \mu_{\theta_m}(\mathbf{x})$ and $\sigma^2_*(\mathbf{x}) = M^{-1} \sum_m \big(\sigma^2_{\theta_m}(\mathbf{x}) + \mu^2_{\theta_m}(\mathbf{x})\big) - \mu^2_*(\mathbf{x})$ respectively.

# 3    Experimental results

## 3.1    Evaluation metrics and experimental setup

For both classification and regression, we evaluate the negative log likelihood (NLL) which depends on the predictive uncertainty. NLL is a proper scoring rule and a popular metric for evaluating predictive uncertainty [49]. For classification we additionally measure classification accuracy and the Brier score, defined as $BS = K^{-1} \sum_{k=1}^{K} \big(t_k^* - p(y = k|\mathbf{x}^*)\big)^2$ where $t_k^* = 1$ if $k = y^*$, and 0 otherwise. For regression problems, we additionally measured the root mean squared error (RMSE). Unless otherwise specified, we used batch size of 100 and Adam optimizer with fixed learning rate of 0.1 in our experiments. We use the same technique for generating adversarial training examples for regression problems. Goodfellow et al. [18] used a fixed $\epsilon$ for all dimensions; this is unsatisfying if the input dimensions have different ranges. Hence, in all of our experiments, we set $\epsilon$ to 0.01 times the range of the training data along that particular dimension. We used the default weight initialization in Torch.

## 3.2    Regression on toy datasets

First, we qualitatively evaluate the performance of the proposed method on a one-dimensional toy regression dataset. This dataset was used by Hernández-Lobato and Adams [24], and consists of 20 training examples drawn as $y = x^3 + \epsilon$ where $\epsilon \sim \mathcal{N}(0, 3^2)$. We used the same architecture as [24].

A commonly used heuristic in practice is to use an ensemble of NNs (trained to minimize MSE), obtain multiple point predictions and use the empirical variance of the networks' predictions as an approximate measure of uncertainty. We demonstrate that this is inferior to learning the variance by training using NLL.[5] The results are shown in Figure 1.

The results clearly demonstrate that (i) learning variance and training using a scoring rule (NLL) leads to improved predictive uncertainty and (ii) ensemble combination improves performance, especially as we move farther from the observed training data.

## 3.3    Regression on real world datasets

In our next experiment, we compare our method to state-of-the-art methods for predictive uncertainty estimation using NNs on regression tasks. We use the experimental setup proposed by Hernández-Lobato and Adams [24] for evaluating probabilistic backpropagation (PBP), which was also used

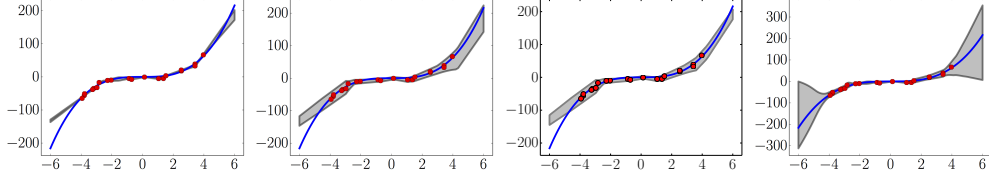

Figure 1: Results on a toy regression task: $x$-axis denotes $x$. On the $y$-axis, the blue line is the *ground truth* curve, the red dots are observed noisy training data points and the gray lines correspond to the predicted mean along with three standard deviations. Left most plot corresponds to empirical variance of 5 networks trained using MSE, second plot shows the effect of training using NLL using a single net, third plot shows the additional effect of adversarial training, and final plot shows the effect of using an ensemble of 5 networks respectively.

by Gal and Ghahramani [15] to evaluate MC-dropout.[6] Each dataset is split into 20 train-test folds, except for the protein dataset which uses 5 folds and the Year Prediction MSD dataset which uses a single train-test split. We use the identical network architecture: 1-hidden layer NN with ReLU nonlinearity [45], containing 50 hidden units for smaller datasets and 100 hidden units for the larger protein and Year Prediction MSD datasets. We trained for 40 epochs; we refer to [24] for further details about the datasets and the experimental protocol. We used 5 networks in our ensemble. Our results are shown in Table 1, along with the PBP and MC-dropout results reported in their respective papers.

| Datasets | RMSE | | | NLL | | |
|---|---|---|---|---|---|---|
| | PBP | MC-dropout | Deep Ensembles | PBP | MC-dropout | Deep Ensembles |
| Boston housing | **3.01 ± 0.18** | **2.97 ± 0.85** | 3.28 ± 1.00 | **2.57 ± 0.09** | **2.46 ± 0.25** | **2.41 ± 0.25** |
| Concrete | **5.67 ± 0.09** | **5.23 ± 0.53** | 6.03 ± 0.58 | **3.16 ± 0.02** | **3.04 ± 0.09** | 3.06 ± 0.18 |
| Energy | **1.80 ± 0.05** | **1.66 ± 0.19** | 2.09 ± 0.29 | 2.04 ± 0.02 | 1.99 ± 0.09 | **1.38 ± 0.22** |
| Kin8nm | 0.10 ± 0.00 | 0.10 ± 0.00 | **0.09 ± 0.00** | -0.90 ± 0.01 | -0.95 ± 0.03 | **-1.20 ± 0.02** |
| Naval propulsion plant | 0.01 ± 0.00 | 0.01 ± 0.00 | **0.00 ± 0.00** | -3.73 ± 0.01 | -3.80 ± 0.05 | **-5.63 ± 0.05** |
| Power plant | **4.12 ± 0.03** | **4.02 ± 0.18** | 4.11 ± 0.17 | 2.84 ± 0.01 | **2.80 ± 0.05** | 2.79 ± 0.04 |
| Protein | 4.73 ± 0.01 | **4.36 ± 0.04** | 4.71 ± 0.06 | 2.97 ± 0.00 | 2.89 ± 0.01 | **2.83 ± 0.02** |
| Wine | **0.64 ± 0.01** | **0.62 ± 0.04** | 0.64 ± 0.04 | 0.97 ± 0.01 | **0.93 ± 0.06** | 0.94 ± 0.12 |
| Yacht | **1.02 ± 0.05** | 1.11 ± 0.38 | 1.58 ± 0.48 | 1.63 ± 0.02 | 1.55 ± 0.12 | **1.18 ± 0.21** |
| Year Prediction MSD | 8.88 ± NA | **8.85 ± NA** | 8.89 ± NA | 3.60 ± NA | 3.59 ± NA | **3.35 ± NA** |

Table 1: Results on regression benchmark datasets comparing RMSE and NLL. See Table 2 for results on variants of our method.

We observe that our method outperforms (or is competitive with) existing methods in terms of NLL. On some datasets, we observe that our method is slightly worse in terms of RMSE. We believe that this is due to the fact that our method optimizes for NLL (which captures predictive uncertainty) instead of MSE. Table 2 in Appendix A.1 reports additional results on variants of our method, demonstrating the advantage of using an ensemble as well as learning variance.

### 3.4 Classification on MNIST, SVHN and ImageNet

Next we evaluate the performance on classification tasks using MNIST and SVHN datasets. Our goal is not to achieve the state-of-the-art performance on these problems, but rather to evaluate the effect of adversarial training as well as the number of networks in the ensemble. To verify if adversarial training helps, we also include a baseline which picks a random signed vector. For MNIST, we used an MLP with 3-hidden layers with 200 hidden units per layer and ReLU non-linearities with batch normalization. For MC-dropout, we added dropout after each non-linearity with 0.1 as the dropout rate.[7] Results are shown in Figure 2(a). We observe that adversarial training and increasing the number of networks in the ensemble significantly improve performance in terms of both classification accuracy as well as NLL and Brier score, illustrating that our method produces well-calibrated uncertainty estimates. Adversarial training leads to better performance than augmenting with random direction. Our method also performs much better than MC-dropout in terms of all the performance measures. Note that augmenting the training dataset with invariances (such as random crop and horizontal flips) is complementary to adversarial training and can potentially improve performance.

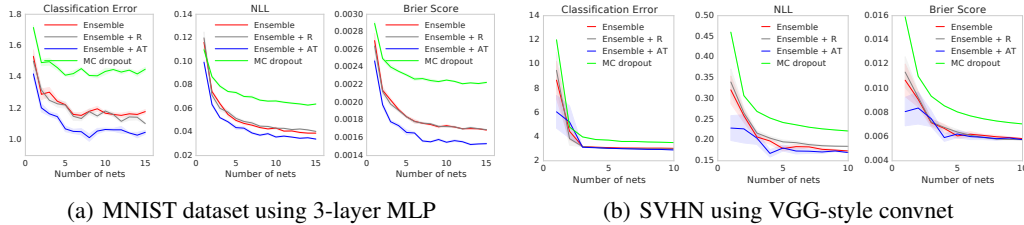

(a) MNIST dataset using 3-layer MLP    (b) SVHN using VGG-style convnet

Figure 2: Evaluating predictive uncertainty as a function of ensemble size $M$ (number of networks in the ensemble or the number of MC-dropout samples): Ensemble variants significantly outperform MC-dropout performance with the corresponding $M$ in terms of all 3 metrics. Adversarial training improves results for MNIST for all $M$ and SVHN when $M = 1$, but the effect drops as $M$ increases.

To measure the sensitivity of the results to the choice of network architecture, we experimented with a two-layer MLP as well as a convolutional NN; we observed qualitatively similar results; see Appendix B.1 in the supplementary material for details.

We also report results on the SVHN dataset using an VGG-style convolutional NN.[8] The results are in Figure 2(b). Ensembles outperform MC dropout. Adversarial training helps slightly for $M = 1$, however the effect drops as the number of networks in the ensemble increases. If the classes are well-separated, adversarial training might not change the classification boundary significantly. It is not clear if this is the case here, further investigation is required.

Finally, we evaluate on the ImageNet (ILSVRC-2012) dataset [51] using the *inception* network [56]. Due to computational constraints, we only evaluate the effect of ensembles on this dataset. The results on ImageNet (single-crop evaluation) are shown in Table 4. We observe that as $M$ increases, both the accuracy and the quality of predictive uncertainty improve significantly.

Another advantage of using an ensemble is that it enables us to easily identify training examples where the individual networks disagree or agree the most. This disagreement[9] provides another useful qualitative way to evaluate predictive uncertainty. Figures 10 and 11 in Appendix B.2 report qualitative evaluation of predictive uncertainty on the MNIST dataset.

## 3.5 Uncertainty evaluation: test examples from known vs unknown classes

In the final experiment, we evaluate uncertainty on out-of-distribution examples from unseen classes. Overconfident predictions on unseen classes pose a challenge for reliable deployment of deep learning models in real world applications. We would like the predictions to exhibit higher uncertainty when the test data is very different from the training data. To test if the proposed method possesses this desirable property, we train a MLP on the standard MNIST train/test split using the same architecture as before. However, in addition to the regular test set with known classes, we also evaluate it on a test set containing unknown classes. We used the test split of the NotMNIST[10] dataset. The images in this dataset have the same size as MNIST, however the labels are alphabets instead of digits. We do not have access to the true conditional probabilities, but we expect the predictions to be closer to uniform on unseen classes compared to the known classes where the predictive probabilities should concentrate on the true targets. We evaluate the entropy of the predictive distribution and use this to evaluate the quality of the uncertainty estimates. The results are shown in Figure 3(a). For known classes (top row), both our method and MC-dropout have low entropy as expected. For unknown classes (bottom row), as $M$ increases, the entropy of deep ensembles increases much faster than MC-dropout indicating that our method is better suited for handling unseen test examples. In particular, MC-dropout seems to give high confidence predictions for some of the test examples, as evidenced by the mode around 0 even for unseen classes. Such overconfident wrong predictions can be problematic in practice when tested on a mixture of known and unknown classes, as we will see in Section 3.6. Comparing different variants of our method, the mode for adversarial training increases slightly faster than the mode for vanilla ensembles indicating that adversarial training is beneficial

for quantifying uncertainty on unseen classes. We qualitatively evaluate results in Figures 12(a) and 12(b) in Appendix B.2. Figure 12(a) shows that the ensemble agreement is highest for letter '*I*' which resembles 1 in the MNIST training dataset, and that the ensemble disagreement is higher for examples visually different from the MNIST training dataset.

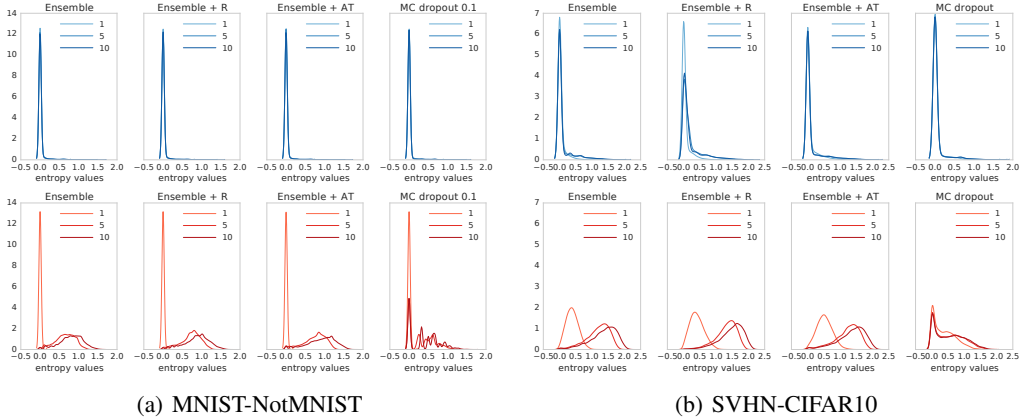

(a) MNIST-NotMNIST                 (b) SVHN-CIFAR10

Figure 3: : Histogram of the predictive entropy on test examples from known classes (top row) and unknown classes (bottom row), as we vary ensemble size $M$.

We ran a similar experiment, training on SVHN and testing on CIFAR-10 [31] test set; both datasets contain $32 \times 32 \times 3$ images, however SVHN contains images of digits whereas CIFAR-10 contains images of object categories. The results are shown in Figure 3(b). As in the MNIST-NotMNIST experiment, we observe that MC-dropout produces over-confident predictions on unseen examples, whereas our method produces higher uncertainty on unseen classes.

Finally, we test on ImageNet by splitting the training set by categories. We split the dataset into images of dogs (known classes) and non-dogs (unknown classes), following Vinyals et al. [58] who proposed this setup for a different task. Figure 5 shows the histogram of the predictive entropy as well as the maximum predicted probability (i.e. confidence in the predicted class). We observe that the predictive uncertainty improves on unseen classes, as the ensemble size increases.

### 3.6 Accuracy as a function of confidence

In practical applications, it is highly desirable for a system to avoid overconfident, incorrect predictions and fail gracefully. To evaluate the usefulness of predictive uncertainty for decision making, we consider a task where the model is evaluated only on cases where the model's confidence is above an user-specified threshold. If the confidence estimates are well-calibrated, one can trust the model's predictions when the reported confidence is high and resort to a different solution (e.g. use human in a loop, or use prediction from a simpler model) when the model is not confident.

We re-use the results from the experiment in the previous section where we trained a network on MNIST and test it on a mix of test examples from MNIST (known classes) and NotMNIST (unknown

| M | Top-1 error % | Top-5 error % | NLL | Brier Score $\times 10^{-3}$ |
|---|---|---|---|---|
| 1 | 22.166 | 6.129 | 0.959 | 0.317 |
| 2 | 20.462 | 5.274 | 0.867 | 0.294 |
| 3 | 19.709 | 4.955 | 0.836 | 0.286 |
| 4 | 19.334 | 4.723 | 0.818 | 0.282 |
| 5 | 19.104 | 4.637 | 0.809 | 0.280 |
| 6 | 18.986 | 4.532 | 0.803 | 0.278 |
| 7 | 18.860 | 4.485 | 0.797 | 0.277 |
| 8 | 18.771 | 4.430 | 0.794 | 0.276 |
| 9 | 18.728 | 4.373 | 0.791 | 0.276 |
| 10 | 18.675 | 4.364 | 0.789 | 0.275 |

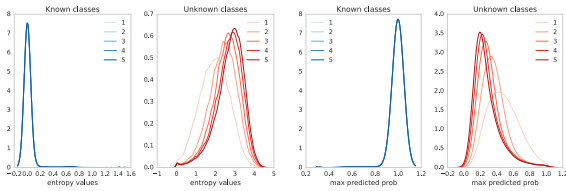

Figure 4: Results on ImageNet: Deep Ensembles lead to lower classification error as well as better predictive uncertainty as evidenced by lower NLL and Brier score.

Figure 5: ImageNet trained only on dogs: Histogram of the predictive entropy (left) and maximum predicted probability (right) on test examples from known classes (dogs) and unknown classes (non-dogs), as we vary the ensemble size.

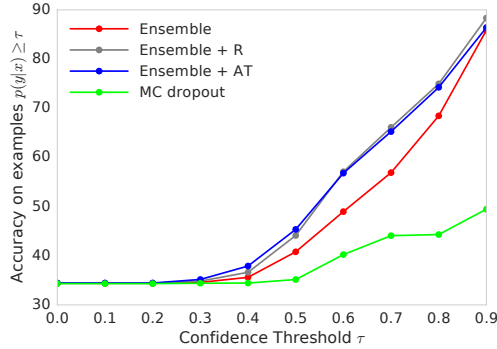

Figure 6: Accuracy vs Confidence curves: Networks trained on MNIST and tested on both MNIST test containing known classes and the NotMNIST dataset containing unseen classes. MC-dropout can produce overconfident wrong predictions, whereas deep ensembles are significantly more robust.

classes). The network will produce incorrect predictions on out-of-distribution examples, however we would like these predictions to have low confidence. Given the prediction $p(y = k|\mathbf{x})$, we define the predicted label as $\hat{y} = \arg\max_k p(y = k|\mathbf{x})$, and the confidence as $p(y = \hat{y}|\mathbf{x}) = \max_k p(y = k|\mathbf{x})$. We filter out test examples, corresponding to a particular confidence threshold $0 \leq \tau \leq 1$ and plot the accuracy for this threshold. The confidence vs accuracy results are shown in Figure 6. If we look at cases only where the confidence is $\geq 90\%$, we expect higher accuracy than cases where confidence $\geq 80\%$, hence the curve should be monotonically increasing. If the application demands an accuracy x%, we can trust the model only in cases where the confidence is greater than the corresponding threshold. Hence, we can compare accuracy of the models for a desired confidence threshold of the application. MC-dropout can produce overconfident wrong predictions as evidenced by low accuracy even for high values of $\tau$, whereas deep ensembles are significantly more robust.

# 4   Discussion

We have proposed a simple and scalable non-Bayesian solution that provides a very strong baseline on evaluation metrics for predictive uncertainty quantification. Intuitively, our method captures two sources of uncertainty. Training a probabilistic NN $p_\theta(y|\mathbf{x})$ using proper scoring rules as training objectives captures ambiguity in targets $y$ for a given $\mathbf{x}$. In addition, our method uses a combination of ensembles (which captures "model uncertainty" by averaging predictions over multiple models consistent with the training data), and adversarial training (which encourages local smoothness), for robustness to model misspecification and out-of-distribution examples. Ensembles, even for $M = 5$, significantly improve uncertainty quality in all the cases. Adversarial training helps on some datasets for some metrics and is not strictly necessary in all cases. Our method requires very little hyperparameter tuning and is well suited for large scale distributed computation and can be readily implemented for a wide variety of architectures such as MLPs, CNNs, etc including those which do not use dropout e.g. residual networks [22]. It is perhaps surprising to the Bayesian deep learning community that a non-Bayesian (yet probabilistic) approach can perform as well as Bayesian NNs. We hope that our work will encourage the community to consider non-Bayesian approaches (such as ensembles) and other interesting evaluation metrics for predictive uncertainty. Concurrent with our work, Hendrycks and Gimpel [23] and Guo et al. [20] have also independently shown that non-Bayesian solutions can produce good predictive uncertainty estimates on some tasks. Abbasi and Gagné [1], Tramèr et al. [57] have also explored ensemble-based solutions to tackle adversarial examples, a particularly hard case of out-of-distribution examples.

There are several avenues for future work. We focused on training independent networks as training can be trivially parallelized. Explicitly de-correlating networks' predictions, e.g. as in [37], might promote ensemble diversity and improve performance even further. Optimizing the ensemble weights, as in stacking [60] or adaptive mixture of experts [28], can further improve the performance. The ensemble has $M$ times more parameters than a single network; for memory-constrained applications, the ensemble can be distilled into a simpler model [10, 26]. It would be also interesting to investigate so-called *implicit ensembles* the where ensemble members share parameters, e.g. using multiple heads [36, 48], snapshot ensembles [27] or swapout [52].

**Acknowledgments**

We would like to thank Samuel Ritter and Oriol Vinyals for help with ImageNet experiments, and Daan Wierstra, David Silver, David Barrett, Ian Osband, Martin Szummer, Peter Dayan, Shakir Mohamed, Theophane Weber, Ulrich Paquet and the anonymous reviewers for helpful feedback.

## Footnotes

[1]Indeed as noted in Gneiting and Raftery [17], it can be shown that asymptotically maximizing any proper scoring rule recovers true parameter values.

[2]We enforce the positivity constraint on the variance by passing the second output through the *softplus* function $\log(1 + \exp(\cdot))$, and add a minimum variance (e.g. $10^{-6}$) for numerical stability.

[3]Not to be confused with Generative Adversarial Networks (GANs).

[4] The bootstrap draws $N$ times uniformly with replacement from a dataset with $N$ items. The probability an item is picked at least once is $1 - (1 - 1/N)^N$, which for large $N$ becomes $1 - e^{-1} \approx 0.632$. Hence, the number of unique data points in a bootstrap sample is $0.632 \times N$ on average.

[5]See also Appendix A.2 for calibration results on a real world dataset.

[6]We do not compare to VI [19] as PBP and MC-dropout outperform VI on these benchmarks.

[7]We also tried dropout rate of 0.5, but that performed worse.

[8]The architecture is similar to the one described in http://torch.ch/blog/2015/07/30/cifar.html.

[9]More precisely, we define disagreement as $\sum_{m=1}^{M} \text{KL}(p_{\theta_m}(y|\mathbf{x})||p_E(y|\mathbf{x}))$ where KL denotes the Kullback-Leibler divergence and $p_E(y|\mathbf{x}) = M^{-1}\sum_m p_{\theta_m}(y|\mathbf{x})$ is the prediction of the ensemble.

[10]Available at http://yaroslavvb.blogspot.co.uk/2011/09/notmnist-dataset.html

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
