[Supplementary Material · supplementary.pdf]

# Supplementary material

## Simple and Scalable Predictive Uncertainty Estimation using Deep Ensembles

## A    Additional results on regression

### A.1    Additional results on regression benchmarks

| Datasets | Ensemble-5 (MSE) | Ensemble-10 (MSE) | ML-1 | ML-1 + AT | ML-5 |
|---|---|---|---|---|---|
| Boston housing | $3.09 \pm 0.84$ | $3.10 \pm 0.83$ | $3.17 \pm 1.00$ | $3.18 \pm 0.99$ | $3.28 \pm 1.00$ |
| Concrete | $5.73 \pm 0.50$ | $5.76 \pm 0.50$ | $6.08 \pm 0.56$ | $6.09 \pm 0.54$ | $6.03 \pm 0.58$ |
| Energy | $1.61 \pm 0.19$ | $1.62 \pm 0.18$ | $2.11 \pm 0.30$ | $2.09 \pm 0.29$ | $2.09 \pm 0.29$ |
| Kin8nm | $0.08 \pm 0.00$ | $0.08 \pm 0.00$ | $0.09 \pm 0.00$ | $0.09 \pm 0.00$ | $0.09 \pm 0.00$ |
| Naval propulsion plant | $0.00 \pm 0.00$ | $0.00 \pm 0.00$ | $0.00 \pm 0.00$ | $0.00 \pm 0.00$ | $0.00 \pm 0.00$ |
| Power plant | $4.08 \pm 0.15$ | $4.07 \pm 0.15$ | $4.10 \pm 0.15$ | $4.10 \pm 0.15$ | $4.11 \pm 0.17$ |
| Protein | $4.49 \pm 0.04$ | $4.50 \pm 0.02$ | $4.64 \pm 0.01$ | $4.75 \pm 0.11$ | $4.71 \pm 0.17$ |
| Wine | $0.64 \pm 0.04$ | $0.64 \pm 0.04$ | $0.64 \pm 0.04$ | $0.64 \pm 0.04$ | $0.64 \pm 0.04$ |
| Yacht | $2.78 \pm 0.59$ | $2.68 \pm 0.57$ | $1.43 \pm 0.57$ | $1.47 \pm 0.58$ | $1.58 \pm 0.48$ |
| Year Prediction MSD | $8.92 \pm$ nan | $8.95 \pm$ nan | $8.89 \pm$ nan | $9.02 \pm$ nan | $8.89 \pm$ nan |

| Datasets | Ensemble-5 (MSE) | Ensemble-10 (MSE) | ML-1 | ML-1 + AT | ML-5 |
|---|---|---|---|---|---|
| Boston housing | $17.28 \pm 6.17$ | $10.61 \pm 4.37$ | $2.55 \pm 0.36$ | $2.57 \pm 0.37$ | $2.41 \pm 0.25$ |
| Concrete | $16.07 \pm 5.75$ | $8.96 \pm 1.73$ | $3.22 \pm 0.31$ | $3.21 \pm 0.26$ | $3.06 \pm 0.18$ |
| Energy | $9.54 \pm 4.54$ | $6.70 \pm 2.39$ | $1.61 \pm 0.40$ | $1.51 \pm 0.28$ | $1.38 \pm 0.22$ |
| Kin8nm | $2.12 \pm 0.97$ | $0.11 \pm 0.20$ | $-1.11 \pm 0.04$ | $-1.12 \pm 0.04$ | $-1.20 \pm 0.02$ |
| Naval propulsion plant | $-5.68 \pm 0.34$ | $-5.85 \pm 0.15$ | $-5.65 \pm 0.28$ | $-4.08 \pm 0.13$ | $-5.63 \pm 0.26$ |
| Power plant | $35.78 \pm 12.87$ | $22.04 \pm 4.42$ | $2.82 \pm 0.04$ | $2.82 \pm 0.04$ | $2.79 \pm 0.04$ |
| Protein | $40.98 \pm 7.43$ | $25.73 \pm 1.59$ | $2.87 \pm 0.03$ | $2.91 \pm 0.03$ | $2.83 \pm 0.02$ |
| Wine | $33.73 \pm 10.75$ | $20.55 \pm 3.72$ | $1.95 \pm 4.08$ | $1.58 \pm 2.30$ | $0.94 \pm 0.12$ |
| Yacht | $10.18 \pm 4.86$ | $6.85 \pm 2.84$ | $1.26 \pm 0.29$ | $1.28 \pm 0.36$ | $1.18 \pm 0.21$ |
| Year Prediction MSD | $39.02 \pm$ nan | $21.45 \pm$ nan | $3.41 \pm$ nan | $3.39 \pm$ nan | $3.35 \pm$ nan |

Table 2: Additional results on regression benchmark datasets: the top table reports RMSE and bottom table reports NLL. Ensemble-$M$ (MSE) denotes ensemble of $M$ networks trained to minimize mean squared error (MSE); the predicted variance is the empirical variance of the individual networks' predictions. ML-1 denotes maximum likelihood with a single network trained to predict the mean and variance as described in Section 2.2.1 . ML-1 is significantly better than Ensemble-5 (MSE) as well as Ensemble-10 (MSE), clearly demonstrating the effect of learning variance. ML-1+AT denotes additional effect of adversarial training (AT); AT does not significantly help on these benchmarks. ML-5, referred to as *deep ensembles* in Table 1, is an ensemble of 5 networks trained to predict mean and variance. ML-5+AT results are very similar to ML-5 (the error bars overlap), hence we do not report them here.

### A.2    Effect of training using MSE vs training using NLL

A commonly used heuristic in practice, is to train an ensemble of NNs to minimize MSE and use the empirical variance of the networks' predictions as an approximate measure of uncertainty. However, this generally does not lead to well-calibrated predictive probabilities as MSE is not a scoring rule that captures predictive uncertainty. As a motivating example, we report calibration curves (also known as reliability diagrams) on the *Year Prediction MSD* dataset in Figure 7. First, we compute the $z\%$ (e.g. $90\%$) prediction interval for each test data point based on Gaussian quantiles using predictive mean and variance. Next, we measure what fraction of test observations fall within this prediction interval. For a well-calibrated regressor, the observed fraction should be close to $z\%$. We compute observed fraction for $z = 10\%$ to $z = 90\%$ in increments of 10. A well-calibrated regressor should lie very close to the diagonal; on the left subplot we observe that the proposed method, which learns the predictive variance, leads to a well-calibrated regressor. However, on the right subplot, we observe that the empirical variance obtained from NNs which do not learn the predictive variance (specifically, five NNs trained to minimize MSE) consistently underestimates the true uncertainty. For instance, the $80\%$ prediction interval contains only $20\%$ of the test observations, which means the empirical variance significantly underestimates the true predictive uncertainty.

Figure 7: Calibration results on the Year Prediction MSD dataset: $x$-axis denotes the expected fraction and $y$-axis denotes the observed fraction; ideal output is the dashed blue line. Predicted variance (left) is significantly better calibrated than the empirical variance (right). See main text for further details.

# B    Additional results on classification

## B.1    Additional results on MNIST

Figures 8 and 9 report results on MNIST dataset using different architecture than those in Figure 2(a). We observe qualitatively similar results. Ensembles outperform MC-dropout and adversarial training improves performance.

Figure 8: Results on MNIST dataset using the same setup as that in Figure 2(a) except that we use two hidden layers in the MLP instead of three. Ensembles and adversarial training improve performance and our method outperforms MC-dropout.

Figure 9:  Results on MNIST dataset using a convolutional network as opposed to the 3-layer MLP in Figure 2(a). Even on a different architecture, ensembles and adversarial training improve performance and our method outperforms MC-dropout.

## B.2    Qualitative evaluation of uncertainty

We qualitatively evaluate the performance of uncertainty in terms of two measures, namely the confidence of the predicted label (i.e. maximum of the softmax output), and the disagreement between the ensemble members in terms of Jensen-Shannon divergence. The results on known classes are

shown in Figures 10 and 11. Similarly, the results on unknown classes are shown in Figures 12(a) and 12(b). We observe that both measures of uncertainty capture meaningful ambiguity.

Figure 10: Results on MNIST showing test examples with high or low disagreement between the networks in the ensemble: Top two rows denote the test examples with least disagreement and the bottom two rows denote test examples with the most disagreement.

Figure 11: Results on MNIST showing test examples with high or low confidence: Top two rows denote the test examples with highest confidence and the bottom two rows denote test examples with the least confidence.

(a) Least and most disagreement    (b) Highest and least confidence

Figure 12: Deep ensemble trained on MNIST and tested on the NotMNIST dataset containing unseen classes: On the left, top two rows denote the test examples with highest confidence and the bottom two rows denote the test examples with the least confidence. On the right, top two rows denote the test examples with highest confidence and the bottom two rows denote the test examples with the least confidence. We observe that these measures capture meaningful ambiguity: for example, the ensemble agreement is high for letters 'I' and some variants of 'J' which resemble 1 in the MNIST training dataset. The ensemble disagreement is high and confidence is low for examples visually dis-similar from the MNIST training dataset.