[Reviews · NeurIPS 2017]

Reviewer 1



The paper investigates the use of an ensemble of neural networks (each modelling the probability of the target given the input and trained with adversarial training) for quantifying predictive uncertainty. A series of experiments shows that this simple approach archives competitive results to the standard Bayesian models. Finding an efficient and reliable way for estimating predictive uncertainty in neural networks its still an open and very interring problem. While the proposed approach is based on well-known models and techniques (and thus is not new itself), to the best of my knowledge it has not been applied to the problem of estimating predictive uncertainty so far and could serve as a good benchmark in future. A drawback compared to the Bayesian models is that the approach comes without mathematical framework and guarantees. Specific comments and questions: - While other approaches, like MC-dropout can also be applied to regression problems with multidimensional targets, it is not clear to me, if a training criterion (like that described in section 2.2.1) suitable for multidimensional targets does also exists (learning a multivariate Gaussian with dependent variables seems not state forward). - "the advantage of VAT is that it does not require knowledge of the true target y and hence can be applied to semi-supervised learning". Note, that adversarial training can also be applied with respect to a different than the correct label (e.g. see https://arxiv.org/pdf/1611.01236.pdf) -"since a base learner trained on a bootstrap sample sees only 63% unique data points". Where are the 63% based on? - "For ease of computing quantiles and predictive probabilities, we further approximate the ensemble prediction as a Gaussian whose mean and variance are respectively the mean and variance of the mixture." Is there a mathematical explanation for why this can be done? - While the likelihood can be estimated directly in the proposed method, it can only be accessed via MCMC methods for MC-dropout. It it nevertheless fair to make the comparison in Table 1? And what do you indicate by the fat printed values? That they are significantly different from each other based on a statistical test? Minor comments: - line 19: natural language processing [36] and bioinformatics [1, 52] -> natural language processing [36], and bioinformatics [1, 52]. - line 61: the aberivation BMA needs to be introduced here. - line 85: and well suited -> and is well suited - line 87: and the first to do so -> and are the first to do so - line 136: the loss is noted differently than before (i.e. in line 108) - line 163: strength and correlation between... -> strength of and correlation between - line 170: additional closing parenthesis. - line 175: helps reduce their bias and ensembling helps reduce the variance -> helps reducing/ helps to reduce - line 194: The Brier score was already defined (differently!) in line 113. - Figure 13 in Abbendix B3: the legend is missing.

Reviewer 2



The paper proposes a simple non-Bayesian baseline method for estimating predictive uncertainty. This is achieved by using ensembles of NNs, where M distinct NNs, from different random initializations, are trained independently through optimizing a scoring rule. The authors simply used a conventional loss function (i.e. softmax cross entropy) as the score rule for classification task, while for regression problems the authors replaced a regular loss function (i.e. MSE) by negative log-likelihood (NLL), which should capture predictive uncertainty estimates. In addition to this, an adversarial training (AT) schema is proposed to promote smoothness over the predictive distribution. To demonstrate the capacity of ensemble and AT to estimate predictive uncertainty, the authors evaluated the proposed method on a large number of datasets across different tasks, both for regression and classification (vision) tasks. Their method outperforms MC dropout for classification tasks while being competitive on regression datasets in term of NLL. Furthermore evaluations demonstrate the ability of the proposed approach to provide uniform predictive distributions for out-of-distribution (unknown) samples, for an object classification task (ImageNet). The paper proposed the use of NNs ensemble, which trained independently on ordinary training samples and their corresponding adversary, in order to estimate the predictive uncertainty. Although the proposed method is not significantly novel, nor very sophisticated, the paper is contributing to the domain by expliciting an idea that have not been very much exploited so far. However, the results obtained from the experimentations are not able to support well the proposal. Training NNs by adversaries is one of the contributions of the paper, but the relationship between smoothness and obtaining a well-calibrated predictive uncertainty estimates is a bit vague. How smoothness obtained using AT can help to achieve a well-calibrated predictive uncertainty estimates? In other words, the effectiveness of AT on the predictive uncertainty estimates is not clearly justified, and the experiments are not supporting this well. For examples, the authors showed AT has no considerable effect for regression problems (Table 2 in Appendix), while it has a significant effect for vision classification problems. Why? Moreover, if we consider AT as data augmentation, can we demonstrate the positive effects of AT on the predictive uncertainty over other simple data augmentation approach, such as random crop? I think that having another experiment with simple data augmentation for training NNs and ConvNets can highlight whether the smoothness has a definite effect on the predictive uncertainty or not, and whether AT is truly key to achieve this. Recognizing erroneous samples, which are confidently misclassified by NNs, is also another concern for AI safety. The high confidence of misclassified samples prohibit a human intervention then some major deficiencies (accidents) can occur. So, the question can be asked that whether this ensemble can recognize the misclassified samples by providing uniform distributions? The experiments should support that. Estimating the predictive uncertainty is a key for recognizing out-of-distribution and misclassified samples. However, a few non-Bayesian papers are missed from the literature review of the paper. For example, Hendrycks and Gimpel (2016) propose a baseline method by using softmax statistics to estimate the probability of error and probability of out-of-distribution samples. Also, ensemble of DNNs have been evaluated by Abbasi and Gagné (2017) for computing the uncertainty of the predictions for adversaries cases, a hard case of out-of-distribution examples. References: Hendrycks, Dan, and Kevin Gimpel (2016). "A baseline for detecting misclassified and out-of-distribution examples in neural networks." arXiv preprint arXiv:1610.02136. Abbasi, Mahdieh, and Christian Gagné (2017). "Robustness to Adversarial Examples through an Ensemble of Specialists." arXiv preprint arXiv:1702.06856. ** Update following reviewers discussions ** I likely underappreciated the paper, re-reading it in the light of other reviews and discussions with other reviewers, I increased my score to 6. I still maintain my comments on AT, which I think is the weakest part of the paper.

Reviewer 3



This paper proposes to use ensembles of deep neural networks to improve estimates of predictive uncertainty in classification and regression. In classification the aggregation is done by averaging probabilistic predictions, obtained by minimising a proper scoring rule, such as cross entropy or Brier score. In regression each component of the ensemble is predicting both mean as well as Gaussian variance around it, and aggregation is done by approximating the mixture of Gaussians by a single Gaussian. The training is enhanced by adversarial training, i.e. creating a shifted copy of each instance, where the shift is adversarially in the direction of largest increase in loss. Extensive experiments demonstrate that this simple method outperforms other methods, including the recently proposed MC dropout by Gal and Ghahramani (ICML 2016). The paper is very well written and a pleasure to read. The experiments are extensive and convincing, the literature review is strong. The only significant shortcoming that I found is that the paper does not discuss running time of the methods. MC dropout and the proposed methods are being compared on the number of dropout samples and number of models in the ensemble, respectively. However, the running time of obtaining a sample from MC dropout should be much faster than training an additional network in the proposed ensemble method. This makes the direct comparisons unfair. However, at least according to the figures, the proposed method still outperforms MC dropout, even if having a smaller ensemble.